# Enduring Effect of Antibiotic Timentin Treatment on Tobacco In Vitro Shoot Growth and Microbiome Diversity

**DOI:** 10.3390/plants11060832

**Published:** 2022-03-21

**Authors:** Inga Tamošiūnė, Elena Andriūnaitė, Jurgita Vinskienė, Vidmantas Stanys, Rytis Rugienius, Danas Baniulis

**Affiliations:** Institute of Horticulture, Lithuanian Research Centre for Agriculture and Forestry, Kaunas Str. 30, LT-54333 Babtai, Kaunas reg., Lithuania; inga.tamosiune@lammc.lt (I.T.); elena.andriunaite@lammc.lt (E.A.); jurgita.vinskiene@lammc.lt (J.V.); vidmantas.stanys@lammc.lt (V.S.); rytis.rugienius@lammc.lt (R.R.)

**Keywords:** 16S rRNA sequencing, *Mycobacterium*, *Nicotiana tabacum*, timentin

## Abstract

Plant in vitro cultures initiated from surface-sterilized explants often harbor complex microbial communities. Antibiotics are commonly used to decontaminate plant tissue culture or during genetic transformation; however, the effect of antibiotic treatment on the diversity of indigenous microbial populations and the consequences on the performance of tissue culture is not completely understood. Therefore, the aim of this study was to assess the effect of antibiotic treatment on the growth and stress level of tobacco (*Nicotiana tabacum* L.) shoots in vitro as well as the composition of the plant-associated microbiome. The study revealed that shoot cultivation on a medium supplemented with 250 mg L^−1^ timentin resulted in 29 ± 4% reduced biomass accumulation and a 1.2–1.6-fold higher level of oxidative stress injury compared to the control samples. Moreover, the growth properties of shoots were only partially restored after transfer to a medium without the antibiotic. Microbiome analysis of the shoot samples using multivariable region-based 16S rRNA gene sequencing revealed a diverse microbial community in the control tobacco shoots, including 59 bacterial families; however, it was largely dominated by *Mycobacteriaceae*. Antibiotic treatment resulted in a decline in microbial diversity (the number of families was reduced 4.5-fold) and increased domination by the *Mycobacteriaceae* family. These results imply that the diversity of the plant-associated microbiome might represent a significant factor contributing to the efficient propagation of in vitro tissue culture.

## 1. Introduction

Endophytes are a class of endosymbiotic microorganisms that internally inhabit plant tissues [1]. Being protected inside explant or seed tissues, endophytic bacteria evade surface sterilization procedures used for cell culture initiation, and they are common in plant tissues grown in vitro [2,3,4]. Some bacterial endophytes maintain habitually concealed lifestyles or, due to limited bacterial growth-supporting media conditions, remain latent over extended periods of in vitro tissue cultivation. In fact, several studies showed a beneficial effect of endophytic bacteria on the growth of in vitro cultures of tomato [5], grapevine [6], sweet cherry [7], apple [8], purple coneflower [9] and tobacco [10]. However, the formation of bacterial colonies on culture media and bacterial overgrowth of plant tissues are common and manifest as infection by a variety of pathogenic species or the non-fastidious proliferation of commensal endophytic bacteria that can be triggered by changes in environmental conditions or plant host physiology [11,12,13].

To eliminate contamination of endophytic origin, pretreatment with high dose [14] or medium supplementation with microbial growth-inhibiting [15] antibiotics have been used and are often reported to improve the propagation or regenerative properties of the tissues [4,16,17,18]. However, antibiotics could be cytotoxic to plant tissues at concentrations required for efficient control of the bacterium, usually in the range of 100 to 500 mg L^−1^ [19] and could lead to tissue browning [20]. Another common application of antibiotics for in vitro tissue culture is related to the use of *Agrobacterium tumefaciens* bacterium-to-plant DNA transfer machinery as an instrument in plant genetic transformation [21]. Although alternative methods for the selection of genetically modified plants have been developed [22], antibiotics remain indispensable for the control of *Agrobacterium* growth upon plant tissue culture transformation. Left untreated, *Agrobacterium*, the causal agent of crown gall disease, shows excessive cell proliferation, which elicits a defense response in plant cells, leading to a detrimental effect on plant tissue growth and efficiency of transformation [23,24,25]. Cephalosporin and penicillin-type antibiotics, such as cefotaxime, carbenicillin and timentin, which are active against the Gram-negative *Agrobacterium* are commonly used [26,27,28]. In culture medium, carbenicillin and penicillins are broken down to physiologically active levels of auxin-related compounds, promoting the organogenesis of explants [29]. However, at higher antibiotic concentrations, proline accumulation, oxidative injury, and reduced antioxidative activity have been reported, which indicates an elevated stress level [30]. Previously, the low cytotoxic effect of the antibiotic timentin on in vitro plant tissues has been described [4,26], and it is commonly used for transformation procedures in the range of concentrations from 150 mg L^−1^ [26] to 500 mg L^−1^ [31], sometimes in combination with lower concentrations of other antibiotics [32].

Cultivated tobacco (*Nicotiana tabacum* L.) has well established in vitro culture conditions [33] and has been commonly used as a model species for genetic transformation studies as well as for practical application in molecular farming using in vitro techniques or contained greenhouse practices [34,35,36]. Recently, the composition of the endophytic microbiome of field-grown tobacco plants or seeds has been explored using cultivation and next-generation sequencing-based methods [37,38] and several endophytic species of the *Bacillus* genus were isolated from tobacco leaves [10,39]. However, the microbial diversity of in vitro cultivated tobacco tissues has not been addressed to date.

Whereas antibiotics are commonly applied for explant and tissue culture decontamination or control of bacterial overgrowth during plant transformation, the effect of antibiotic treatment on the composition of endogenous microbial population in tissue culture and the long-term consequences on plant response to stress and growth parameters have not been completely understood. Considering the established role of endophytic bacteria as modulators of plant growth and adaptation, it could be expected that antibiotic-induced perturbation of the endophytic community might affect the efficiency of in vitro culture propagation. Therefore, the aim of this study was to assess the effect of antibiotic timentin, commonly used following *Agrobacterium*-mediated transformation, on the growth and stress level of tobacco shoots in vitro and the composition of the plant-associated microbiome. We assessed changes in shoot biomass accumulation and oxidative stress injury of cellular membranes during the antibiotic treatment and residual effect following the treatment. We also investigated the composition of the shoot bacterial population before and after antibiotic treatment using a multivariable region-based 16S rRNA gene high-throughput sequencing analysis. To further refine the taxonomic assignment of the dominant taxa that was revealed by the microbiome analysis, we cultured a *Mycobacterium* sp. isolate and tested its antibiotic resistance profile.

## 2. Results

### 2.1. Effect of Antibiotic on Tobacco Shoot Growth and Oxidative Stress Injury

Supplementation of growth medium with 250 mg L^−1^ timentin had a growth-suppressing effect on tobacco in vitro shoot culture. After three weeks of cultivation, the average fresh weight (FW) of the control tobacco shoots (TC) was estimated to be 164.7 ± 3.3 mg (Figure 1, Appendix B
Table A1). The FW of the shoots cultivated on medium containing timentin (TA) was reduced by 29 ± 4% (*p* = 1.5 10^−7^) compared to the TC shoots.

An analysis of the membrane lipid oxidative injury of the TC shoots revealed significant variation in malondialdehyde (MDA) concentration during the three weeks of the propagation cycle (Figure 2; Appendix B
Table A1). The highest value was detected at day four after transfer to fresh medium and was estimated at 95.6 ± 2.6 nmol g^−1^ FW. The increased MDA accumulation in the control shoots was likely a consequence of the combined effect of stress associated with tissue senescence and shoot injury during the transfer to the fresh medium, which was followed by adaptation and active growth on the fresh medium. This resulted in an ~1.3-fold (*p* = 4 × 10^−6^) decrease in MDA concentration at the end of the first week of cultivation (day 7) and a subsequent gradual increase in MDA concentration during the course of culture senescence over the remaining two weeks of the propagation cycle. However, the TA shoots maintained on the antibiotic supplemented medium did not follow this cycle and showed 1.2–1.6-fold (*p* < 0.014) higher levels of MDA than the TC shoots over the entire period of the propagation cycle (Figure 2), which could be a direct consequence of antibiotic-induced cytotoxic effects.

Interestingly, after transfer to the medium without antibiotic (PA), a consistent residual negative effect of antibiotic treatment on shoot growth vigor and stress level was observed for at least several passages used in the experiments. Cumulative data from several of these experiments revealed that the growth vigor of the shoots was only partially restored, and that the accumulation of biomass remained significantly lower (9 ± 3%, *p* = 0.013) compared to TC (Appendix B
Table A1). Similarly, the level of MDA accumulation was reduced compared with TA and showed a significant decrease during the first week after transfer to fresh medium. Nevertheless, it remained 1.1–1.3-fold (*p* < 0.033) higher compared to the control group over the remaining two weeks of the propagation cycle.

### 2.2. Analysis of Bacterial Diversity in Tobacco In Vitro Shoot Samples

To investigate bacterial diversity in tobacco shoot culture and to assess the effect of antibiotic treatment on bacterial diversity, DNA extracted from shoots of the TC and PA experimental groups was subjected to 16S rRNA gene amplicon analysis using the Ion Torrent high-throughput sequencing platform.

The overall number of high-quality mapped sequences with 224–225 bp read length was similar for the TC and PA libraries (1,027,576 and 1,175,938, respectively) but varied up to ~3.5-fold for different DNA preparation methods and primer pairs specific to distinct 16S rRNA gene regions (Figure 3, Appendix B
Table A2). The proportion of bacterial sequences varied from 10% to 29% for the different DNA extraction methods (Figure 3a). Independent of the experimental group, consistently higher content of bacterial operational taxonomic units (OTUs) (>25% of all reads) was detected for samples prepared with DNA extraction methods described by Doyle [40] and Ding et al. [41]. For the TC shoots, similar results were also obtained using the DNA extraction method described by Li et al. [42]. The remaining two methods resulted in a consistently lower proportion of the bacterial OTUs. Bacterial DNA enrichment by selective organelle lysis using SDS and NaCl as previously described by Wang et al. [43] for preparation of microbial metagenomic libraries of tropic tree *Mallotus nudiflorus* did not increase bacterial OTU content for the tobacco shoot samples as compared to other DNA extraction methods, which suggests that this approach requires further optimization.

The proportion of sequences assigned to the bacterial and tobacco plastid or mitochondrial 16S rRNA also varied depending on the PCR amplification region and/or specificity. Two primer pairs specific to the V4 and V8 regions generated the largest number of reads mapped as bacterial OTUs and the largest proportion of the bacterial OTUs compared to the overall number of reads (Figure 3b). Meanwhile, the overall amplification efficiency of the V6–7 and V2 primers was 2- to 4-fold lower, and the resulting proportion of bacterial sequences was comparable to that of V8. The V3 primers generated a comparable number of reads in total to the V8 region, but these reads contained the lowest proportion of bacterial sequences. Notably, for all primer sets, a consistently lower ratio of bacterial to plastid/mitochondrial OTUs was obtained for the PA sample than for the TC sample which could be a result of lower yields of bacterial DNA due to lower bacterial density in the antibiotic-treated shoot sample.

The 16S rRNA region-specific primer propensity for specific taxonomic groups was assessed using family-level data to avoid bias due to the limited accuracy of genus or species identification using short-read sequences which could lead to underestimation of primer specificity. The association among the OTUs obtained using different primer pairs was mapped on the UpSet plot using cumulative data from both experimental groups (Figure 4). Among the 59 family-level OTUs, 22% were detected by all primer pairs and were represented by 96% or 70% of all bacterial reads when the dominant *Mycobacteriaceae* family was included or excluded from the analysis, respectively. No singleton families were detected using V2-specific primers and 26 (42% of all families) singleton families were detected by the remaining four primer pairs.

Previously, variation in the informative power of the same 16S rRNA region-specific primers has been demonstrated in applications to human intestinal and environmental microbiota samples [44,45,46]. In our study, the read abundance for the same OTU also varied considerably among the primer pairs (Figure 5). For example, the most abundant *Mycobacteriaceae* family was represented by 37%, 18%, and 39% of the total number of reads using V4, V6–7 and V8 primers, while only 6% and <1% of the total number of reads were assigned to the family for the V2 and V3 primer data sets, respectively. In another instance, the distribution of reads assigned to the *Paenibacillaceae* family among the V3, V4, and V8 data sets was 15%, 38%, and 44%, respectively, while V2 and V6–7 represented only 3% and <1% of the total number of reads, respectively. Regardless of the apparent taxa-specific amplification efficiency among the primer pairs, the hierarchical cluster analysis of the data did not reveal a consistent distribution of taxonomic groups among the primer data sets (Figure 5), possibly due to variation of primer specificity at a lower taxonomic level or the bias introduced by variation of the taxa abundance.

### 2.3. Antibiotic Effect on Bacterial Diversity in Tobacco Shoot Culture

Although both the TA and PA samples had a similar overall number of reads and reads mapped to bacterial OTUs, the number of unique OTUs was reduced from 153 to 27 upon antibiotic treatment (Appendix B
Table A2). Beta-diversity analysis of the microbiome data sets using NMDS with Bray–Curtis dissimilarity matrix represented variation between the two tobacco shoot experimental groups on the first coordinate (Figure 6).

Among the six phyla detected in the tobacco shoot microbiome, *Actinobacteria* was dominant in both experimental groups and was mainly represented by the order *Actinomycetales*, including 17 and 3 families for the TC and PA experimental groups, respectively (indicated in a blue font in Figure 5; Appendix A). *Actinobacteria* included the most prevalent family in the data set, *Mycobacteriaceae*, which represented 81% and 98% of mapped bacterial reads for TC and PA, respectively. *Microbacteriaceae* represented another antibiotic treatment-enduring *Actinobacteria* that became the second most abundant OTU with a relative abundance of 1.6% in the antibiotic-treated shoot sample, while *Propionibacteriaceae* and *Acidimicrobiaceae* were detected only at marginal levels. For OTUs in the latter family, sequence comparison showed 98% sequence similarity to *Mycobacteriaceae*, which could also imply inaccuracy of the sequence assignment. A similar assumption could be drawn for the assignment of the *Thermolithobacteraceae* family of the phylum *Firmicutes*. Meanwhile, *Bacillaceae* and *Paenibacillaceae* were the most abundant among the remaining four families of *Firmicutes* representing 3% of mapped bacterial reads in the control shoots. In addition, *Staphylococcaceae* and *Streptococcaceae* were detected at low abundance but by three primer pairs each. The abundance of all *Firmicutes* was largely reduced (<0.1% of mapped reads) upon antibiotic treatment.

In the control shoots, 1.6% of mapped reads were assigned to five families of *Bacteriodetes*, mainly represented by *Sphingobacteriaceae*, *Chitinophagaceae*, and *Flavobacteriaceae*, but all of them were undetectable in the antibiotic-treated sample. Among the *Proteobacteria* including 8% and 0.5% of mapped bacterial reads of the TC and PA, respectively, class *Alpha-proteobacteria* included 10 families of order *Rhizobiales* representing many well-known beneficial plant-associated bacteria [47] and *Caulobacteraceae* and *Sphingomonadaceae* families including common environmental bacteria [48,49]. In the antibiotic-treated shoots only *Caulobacteraceae* and *Bradyrhizobiaceae* were detected at 4- and 10-fold reduced abundance levels compared to control shoots, respectively. Interestingly, the families *Kopriimonadaceae*, *Rhodobacteraceae,* and *Rhodospirillaceae*, which includes species of common environmental and aquatic bacteria, were detected at low abundance only in the antibiotic-treated shoots.

The *Desulfovibrionaceae* family of class *Delta-proteobacteria* was detected at low but similar abundance in both TC and PA shoots. However, more proliferous *Beta*-*proteobacteria* and *Gamma*-*proteobacteria* (including 5 and 7 families, respectively) were also largely reduced upon antibiotic treatment. Among the *Beta*-*proteobacteria* were notable *Alcaligenaceae* and *Methylophilaceae* which include nitrifying and methylotrophic environmental bacteria [50,51]. Meanwhile, the *Enterobacteriaceae* and *Pseudomonadaceae* families of *Gamma-proteobacteria* represent numerous plant endophytes or pathogens [52]. In addition, two families of the PVC superphylum, *Planctomycetes* and *Verrucomicrobia*, which include common soil and plant root-associated bacteria [53] were both detected at low abundance in the control shoots.

### 2.4. Isolation of Actinobacteria and Antibiotic Resistance Assessment

To further characterize the *Actinobacteria* dominant in the in vitro tobacco shoot culture, an extract of antibiotic-treated shoots was plated on Loewenstein–Jensen growth medium. The isolate obtained from bright yellow colonies formed after 6–8 weeks of incubation showed 98% identity to the *M. cookii* strain ATCC 49103 (GenBank accession NR_114661.1) [54] based on the 1407 nt fragment of 16S rRNA. The isolate also sustained similar growth properties on Actinobacteria Isolation Agar, which was used for later cultivation and antibiotic resistance tests. The isolate showed resistance to timentin at the concentration used for the tobacco shoot treatment (250 mg L^−1^) and chloramphenicol at 30 mg L^−1^ (Figure 7); however, the growth of the isolate was suppressed by rifampicin at 25 mg L^−1^, which suggests this antibiotic could be used to eliminate mycobacteria from tobacco in vitro cultures.

## 3. Discussion

Our study confirmed that application of multivariable region-based approach for 16S rRNA gene high-throughput sequencing can provide higher taxonomic resolution of microbial diversity compared to results obtained using individual regions. However, five primer pairs of the 16S Metagenomics Kit used in the analysis showed different specificity for bacterial and tobacco plastid or mitochondrial 16S rRNA sequences. Despite the fact that the V3 region was previously shown to efficiently represent the diversity of microbial communities of fecal or sewage samples [55,56], our analysis using plant-derived samples revealed that V3 had the strongest predisposition toward plastid and/or mitochondrial sequences.

An abundant microbial community of the control tobacco shoots was revealed by the 16S rRNA gene high-throughput sequencing analysis, and 153 detected OTUs were assigned to 59 families of the Bacteria domain (Figure 5). Bacterial endophytes are common within in vitro cultures initiated from surface-sterilized plant dormant tissues, such as seeds or buds, which contain a complex community of endophytic bacteria. A prior study by Thomas et al. [57] revealed the vast diversity of endophytic bacteria prevailing in grapevine field shoots (mainly *Proteobacteria* but also *Actinobacteria*, *Firmicutes* and several less abundant phyla) and their introduction to in vitro shoot culture was demonstrated. The occurrence of endophytic bacteria in seeds is also well documented [58,59]. Since the tobacco in vitro cultures used in our study were initiated from surface-sterilized seeds, it is possible that numerous bacterial species survived the sterilization procedure in internal tissues or even on the surface of seeds. Endophytic species vertically transmitted through seeds are closely associated with plant hosts and often play a significant role in the regulation of plant growth and developmental physiology. Previously, *Enterobacteriaceae* was detected as the predominant species of the endophyte community of tobacco seeds of the four distinct tobacco cultivars, and a genotype-specific signature was observed for *Alpha-proteobacteria* [38]. In our study, among the bacteria identified in the shoot microbiome, numerous species of *Bacillales*, *Rhizobiales*, *Burkholderiales,* or *Enterobacteriales* could include plant endophytic bacteria adapted for vertical transmission and capable of inhabiting seed tissues.

Alternatively, endospore- or biofilm-forming bacteria are adapted to survive in extreme environments, and many of the bacteria show resilience to sterilizing agents and would endure seed sterilization procedures. Such a route could be adequate for soil bacteria such as *Mycobacterium* spp. which form resilient endospores [60], are capable of colonizing plant tissues but have not been detected in seeds so far [61,62]. Similarly, species of *Sphingomonadaceae* are widely distributed in nature, have been isolated from many different land and water habitats, as well as from plant root systems, and can form resilient biofilms that are often capable of surviving sterilization conditions [63].

The control tobacco shoot microbiome was largely dominated by *Mycobacteriaceae*, and the isolate closely related to *Mycobacterium cookii* was likely among the dominant species. Mycobacteria are widely distributed in water and soil [64]; however, although the genus *Mycobacterium* comprises nearly 200 species, to date, research has mainly focused on obligate pathogens and information about plant-associated mycobacteria is relatively scarce. *Mycobacterium* spp. have been isolated from the rhizosphere of tomato [65] and rice plants [66]. They have also been detected by sequencing-based analysis as endophytes of rice roots and stems [62], wheat roots [67], shoots of rock plant [68], and buds of Scots pine [69]. Endophytic *Mycobacterium* isolated from orchid plants was shown to stimulate seed germination and plant growth [70,71,72]. Although *Mycobacterium* spp. have not previously been described as tobacco endophytes, they could be, particularly depending on cultivation conditions. Specifically, endophytic, or epiphytic colonization would be plausible for the plants cultivated on peat substrate considering that mycobacteria are abundant in peat as they are common inhabitants of sphagnum vegetation [73], which was also one of the first described habitats for *M. cookii* [74]. It is notable that previously *Mycobacterium* spp. have been detected as dominant species in sweet cherry in vitro shoots [75] and tissue cultures of Scots pine [76].

The low cytotoxic effect of the antibiotic timentin on in vitro plant tissues had been documented [4,26,31], and it had been commonly used for plant cell culture [26,31]. However, our experiments revealed suppressed growth of tobacco in vitro shoots on a medium supplemented with timentin and an enduring negative effect of antibiotic treatment on shoot growth vigor and stress level was observed after transfer to the medium without the antibiotic. Microbiome analysis revealed that treatment with the antibiotic had a long-term detrimental effect on the endophytic community of tobacco in vitro shoots. The number of detected families was reduced ~4.5-fold and most of the remaining families had largely reduced relative abundances. Interestingly, antibiotic treatment had little effect on the abundance of the dominant *Mycobacteriaceae* and *Microbacteriaceae* families (Figure 5, Appendix A) as well as two *Alpha-proteobacteria* families, *Caulobacteraceae* and *Bradyrhizobiaceae*, which showed a modest reduction in relative abundance. Recently, the antibiotic effect on the diversity and structure of bacterial communities was described in soil samples [77]. The study revealed that tetracycline addition could change the microbial community composition and relative abundance of bacteria due to a combination of growth suppressing or bactericidal effect of tetracycline and the induction of bacterial antibiotic resistance. In our study, the antibiotic resistance of the *M. cookii*-related isolate supported the notion that increased domination of the *Mycobacteriaceae* family in the timentin-treated shoots is a consequence of antibiotic resistance. Resistance to carboxypenicillins, such as ticarcillin, has been well documented for a variety of bacteria including actinobacterial taxa [78]. In addition, resistance to clavulanic acid inhibitors would be required to ensure effective growth observed for the *Mycobacterium* isolate, which is relatively common among bacteria [79]. Resistance to antibiotics is also the most likely explanation for the unchanged or moderately decreased abundance of other bacterial taxa detected in the antibiotic-treated shoots; however, this would require further experimental confirmation.

Although some species of *Mycobacterium* have been reported to suppress undifferentiated Scots pine tissue growth and reduce the length of hypocotyls of seedlings, no negative effect has been observed for other species [76]. However, so far, plant growth regulating properties have not been described for species related to the *M. cookii* isolated in our study.

Antibiotic-induced perturbations in composition and/or interactions within the plant-associated microbial community of the tobacco shoots could be a significant factor contributing to the enduring negative effect on the growth and adaptive capacity of the plant tissue culture. The plant microbial community has a profound effect on the health of the host plant, and reduced microbial diversity often leads to disease development [80,81]. It could be proposed that antibiotic-induced perturbations in microbial community structure could eliminate pathogen-suppressing antagonistic interactions and facilitate the spread of pathogenic microorganisms or that the disease susceptibility of the plants could be increased due to elevated stress levels. However, the antibiotic-treated tobacco shoots did not show symptoms that could be attributed to microbial pathogenesis, such as distorted morphology or necrosis of tissues (Figure 1). Alternatively, a loss of mutualistic interaction with beneficial endophytic bacteria could contribute to reduced adaptation to in vitro conditions and inhibited shoot growth. Indeed, growth-promoting [5,9] stress-reducing activities [6,8,10] of bacteria on in vitro propagated plants have been reported previously. Our study revealed that antibiotic treatment significantly reduced the abundance of several families of order Rhizobiales as well as *Caulobacteraceae* and *Sphingomonadaceae* that involve many common environmental bacteria including beneficial plant-associated bacteria [47,48,49]. The treatment also had a negative effect on the abundance of nitrifying and methylotrophic environmental bacteria of *Alcaligenaceae* and *Methylophilaceae* [50,51] as well as *Enterobacteriaceae* and *Pseudomonadaceae* representing numerous plant endophytes as well as pathogens [52]. The large diversity of bacteria affected by the antibiotic treatment detected in our analysis would lead to an extensive list of potential candidates for beneficial interactions; therefore, further studies, potentially including specific bacterial isolates or microbial consortia, would be required to refine the role of the plant microbiome in the modulation of in vitro culture growth and adaptation to the in vitro environment.

## 4. Materials and Methods

### 4.1. Tobacco Shoot Culture In Vitro and Antibiotic Treatment

Cultivated tobacco (*Nicotiana tabacum* cv. Samsun-NN) in vitro shoot culture was derived from seeds. Tobacco shoots cultivated in vitro for over five years were used as a control (TC) and were maintained on solid Murashige–Skoog (MS) medium [82] supplemented with 0.75 mg L^−1^ 6-benzylaminopurine, 30 g L^−1^ sucrose, 0.8% agar at 25 ± 1 °C under 150 μmol m^−2^ s^−1^ intensity illumination with a 16 h photoperiod. Antibiotic-treated tobacco shoots (TA) were maintained as described for the TC, except the medium was supplemented with timentin at 250 mg L^−1^. To ascertain a stable and homogeneous physiological response in the shoot culture, the duration of the treatment was 6 months. Prolonged cultivation was also used to clear residual DNA from antibiotic-inactivated bacteria to avoid false-positive results during the DNA sequencing-based shoot microbiome analysis. After transfer to medium without antibiotic (PA), the shoots were maintained under the same conditions as TC and at least one culture passage (one-month duration) was used before collecting samples for the analysis.

Samples for shoot fresh weight (FW) and microbial composition analysis were collected three weeks after transfer to fresh medium, and oxidative stress injury was assessed at four distinct time points during the propagation cycle.

### 4.2. Assessment of Oxidative Stress Injury

Oxidative injury of tobacco shoot cellular membranes was estimated based on quantitative analysis of the lipid peroxidation product malondialdehyde (MDA) using a previously described method [83,84]. Homogenized frozen tobacco shoot powder was extracted with 50 mM Tris-HCl pH 7.4 containing 1.5% polyvinylpolypyrrolidone for 30 min at 4 °C and centrifuged at 10,000× *g* for 15 min at 4 °C. Equal amounts of tissue extract and 0.5% thiobarbituric acid in 20% trichloroacetic acid were mixed, heated at 95 °C for 30 min, cooled on ice, and centrifuged at 10,000× *g* for 5 min. The absorbance measured at 532 nm was corrected by subtracting the absorbance value at 600 nm and MDA concentration was estimated using ε = 155 mM^−1^ cm^−1^. The absence of interference from the absorbance of anthocyanins at 532 nm was verified using control samples without thiobarbituric acid.

### 4.3. 16S rRNA Gene Sequencing-Based Microbiome Analysis

The samples from the TC and PA experimental groups were fresh frozen in liquid N_2_ and stored at −70 °C. To assess which methods are most efficient for bacterial DNA extraction from the plastid and mitochondrial DNA-rich plant material, four different DNA extraction methods described by [40,41,85,86], PureLink Microbiome DNA Purification kit (Thermo Fisher Scientific, Waltham, MA, USA) and a method for bacterial DNA enrichment using SDS detergent extraction [43] were used for DNA preparation from independent samples including 15 to 20 tobacco shoots. The PCR amplicon libraries were generated using a multivariable region approach comprising six regions of the 16S rRNA represented by a combination of five primer pairs (V2, V3, V4, V6–7, and V8) included in the 16S Metagenomic kit (Thermo-Fisher Scientific, Waltham, MA, USA) (Figure 8) as described previously [87]. Equal volumes of all DNA library samples (adjusted to 10 pM) were combined and emulsion PCR was carried out using Ion OneTouch 2 System and Ion PGM Hi-Q View OT2 Kit (Thermo Fisher Scientific, Waltham, MA, USA). The clonal libraries were enriched using Ion OneTouch ES (Thermo Fisher Scientific, Waltham, MA, USA) and sequencing was performed using an Ion 316 v.2 chip on the Ion Personal Genome Machine system using an Ion PGM Hi-Q Sequencing kit (Thermo Fisher Scientific, Waltham, MA, USA). Base calling and run demultiplexing were performed by Torrent Suite v.5.0.5 (Thermo Fisher Scientific, Waltham, MA, USA) with default parameters. Sequencing data were processed using the 16S Metagenomic workflow of Ion Reporter Software v.5.10.5.0 (Thermo Fisher Scientific, Waltham, MA, USA). Primers were trimmed from the reads at both ends. The threshold for unique reads was set to 10. Taxonomic identification was performed using MicroSEQ 16S Reference Library v.2013.1 and the Greengenes v.13.5 databases. N. tabacum plastid 16S rRNA (NC_001879.2:102762-104252) and mitochondrion 18S rRNA (NC_006581.1:108334-110235) sequences were also included to detect sequences resulting from contamination of the plant mitochondrial and plastid host DNA. The threshold values for percentage identity for genus and species IDs were 97% and 99%, respectively.

### 4.4. Mycobacterium Isolation, Identification, and Antibiotic Resistance Test

Tobacco shoots were homogenized with a razor blade in MS medium, the homogenate was applied to Loewenstein–Jensen medium prepared from TB-Medium base (Merck, Darmstadt, Germany) and incubated at room temperature for 6–8 weeks. The bacterial isolate was maintained on Actinobacteria Isolation Agar (AIA). Bacterial DNA was isolated using the GeneJET Genomic DNA Purification kit (Thermo Fisher Scientific, Waltham, MA, USA). The 16S rRNA gene fragment was amplified using universal primers as described previously [10] and the sequence was queried at the NCBI BLAST server [88]. Isolate resistance to antibiotics was tested by cultivation on AIA medium supplemented with timentin (250 mg L^−1^), chloramphenicol (30 mg L^−1^), or rifampicin (25 mg L^−1^).

### 4.5. Statistical Data Analysis

Statistically significant differences in shoot FW and MDA accumulation were assessed by ANOVA analysis and Tukey’s post hoc test using IBM SPSS Statistics v.21 (IBM Inc., Armonk, NY, USA). Data are presented as the mean of at least 3 independent experiments and standard error of the mean.

The 16S rRNA gene sequencing data were rarefied to the minimum library size and total sum normalization was applied to taxonomic count data by dividing feature read counts by the total number of reads in each sample. Nonmetric multidimensional scaling (NMDS) was performed using the Microbiome Analyst [89] and HeatMap function of the Calypso web-based platform was used for visualization of the OTU distribution [90]. The representation of the microbiome by the primer pairs representing distinct rRNA regions was visualized using UpSet graph [91].

## 5. Conclusions

Our study revealed that the antibiotic timentin-induced suppression of growth and elevated stress level of tobacco in vitro shoots was followed by an enduring residual negative effect after shoot transfer to medium without antibiotics. The analysis of the control tobacco shoot microbiome detected an extensive bacterial community dominated by the *Mycobacteriaceae* family, which potentially could originate from seeds used to initiate the in vitro culture and might include endophytic species closely associated with the plant host as well as contamination of resilient environmental bacteria. The antibiotic treatment-induced decline in taxonomic diversity and increased dominance of *Mycobacteriaceae* and several other families was likely associated with antibiotic resistance traits of the bacteria. It was proposed that antibiotic-induced perturbation of shoot microbiome composition and/or interactions might contribute to the reduced adaptive capacity and impede the growth of tobacco shoots, leading to the reduced efficiency of in vitro culture propagation. Therefore, further insights into the specific role of the members and interactions of the tobacco shoot microbiome would ensure a better understanding of the antibiotic effect on the in vitro tissue and would potentially provide solutions required to improve the efficacy of the in vitro tissue culture.

## Figures and Tables

**Figure 1 plants-11-00832-f001:**
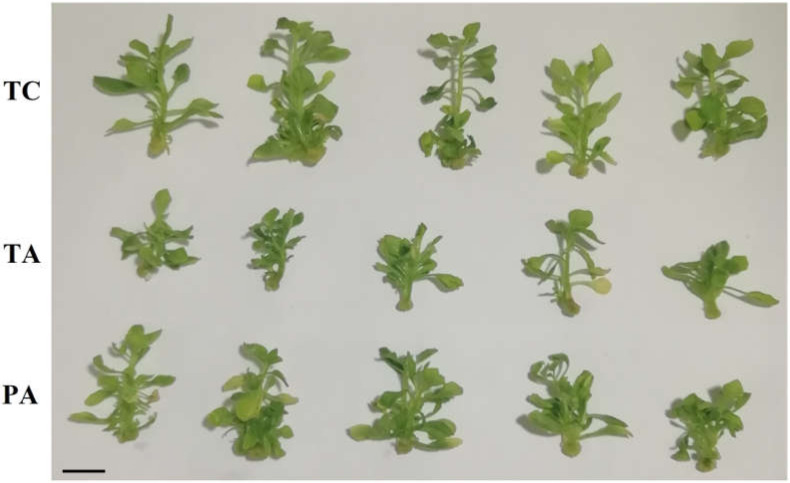
Representative image of control (TC), 250 mg L^−1^ timentin-treated (TA) and post-antibiotic treatment (PA) tobacco in vitro shoots cultivated for 3 weeks. The scale bar size represents 10 mm.

**Figure 2 plants-11-00832-f002:**
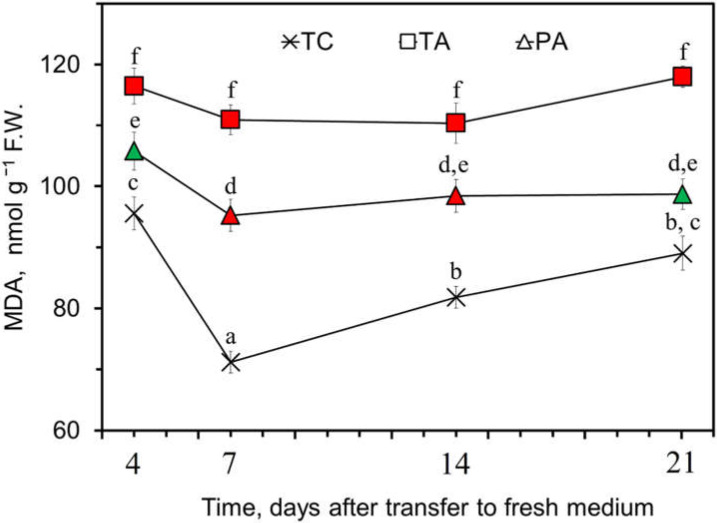
Effect of antibiotic treatment on MDA accumulation in tobacco in vitro shoots. Control (TC) (×), 250 mg L^−1^ timentin-treated (TA) (**□**) and post-antibiotic treatment (PA) (∆) experimental groups were maintained as described in the Methods. Data from four independent experiments are presented as the mean ± standard error (Appendix B
Table A1). The color of the symbols represents significant differences of the mean values compared to the control at each time point (green—*p* < 0.05, red—*p* < 0.01). Different letters denote significant differences among the time points (*p* < 0.05).

**Figure 3 plants-11-00832-f003:**
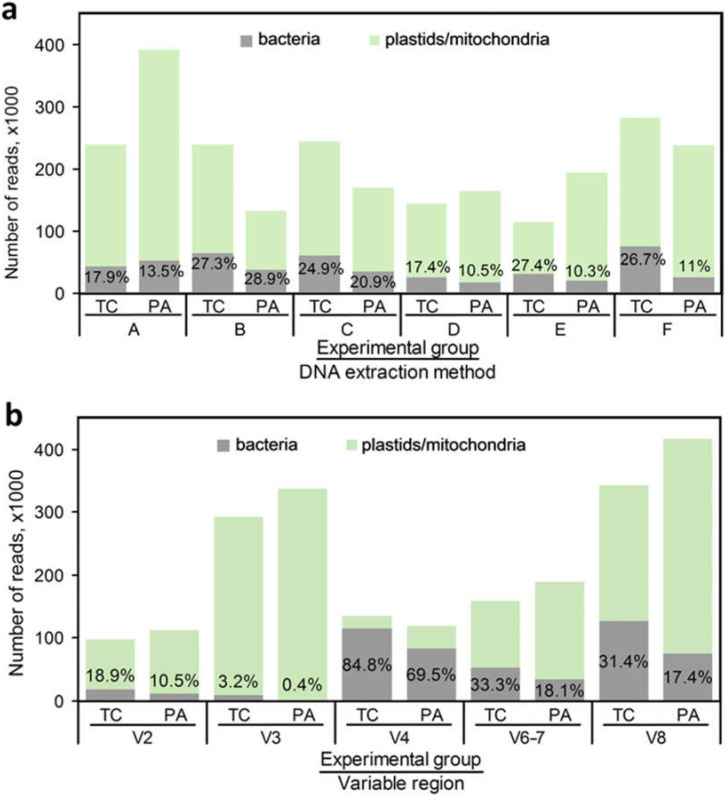
Distribution of the overall number of mapped reads and proportion of bacterial OTUs (blue) to plastid or mitochondrial 16S rRNA sequences (green) generated for the tobacco shoot control (TC) and post-antibiotic treatment (PA) samples using six different DNA extraction and enrichment methods (A—Akinsanya et al., 2015; B—Doyle, 1991; C—Ding et al., 2013; D—PureLink Microbiome DNA Purification Kit (Thermo-Fisher Scientific); E—Li et al., 2001; F—Wang et al., 2008) (**a**) or five primer pairs specific to the V2, V3, V4, V6–7 and V8 variable regions of 16S rRNA gene (**b**). Numbers indicate the percent value of reads mapped to bacterial sequences estimated based on the total number of mapped reads for each individual sample.

**Figure 4 plants-11-00832-f004:**
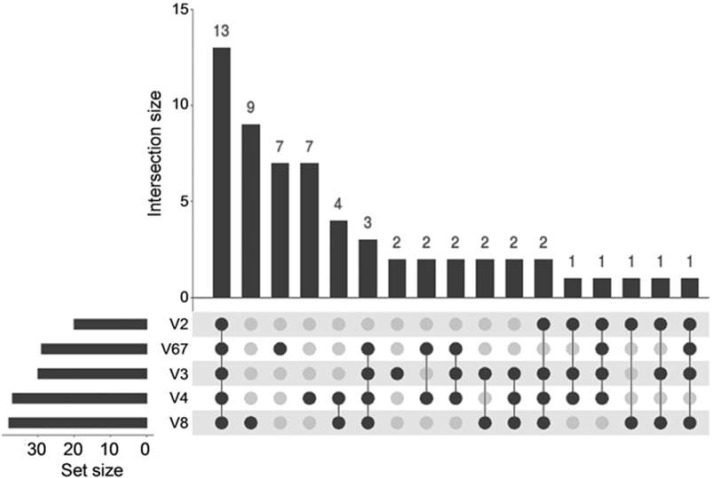
Association among the five 16S rRNA variable region-specific primer pairs estimated using combined family-level data of the tobacco shoot control (TC) and post-antibiotic treatment (PA) experimental groups.

**Figure 5 plants-11-00832-f005:**
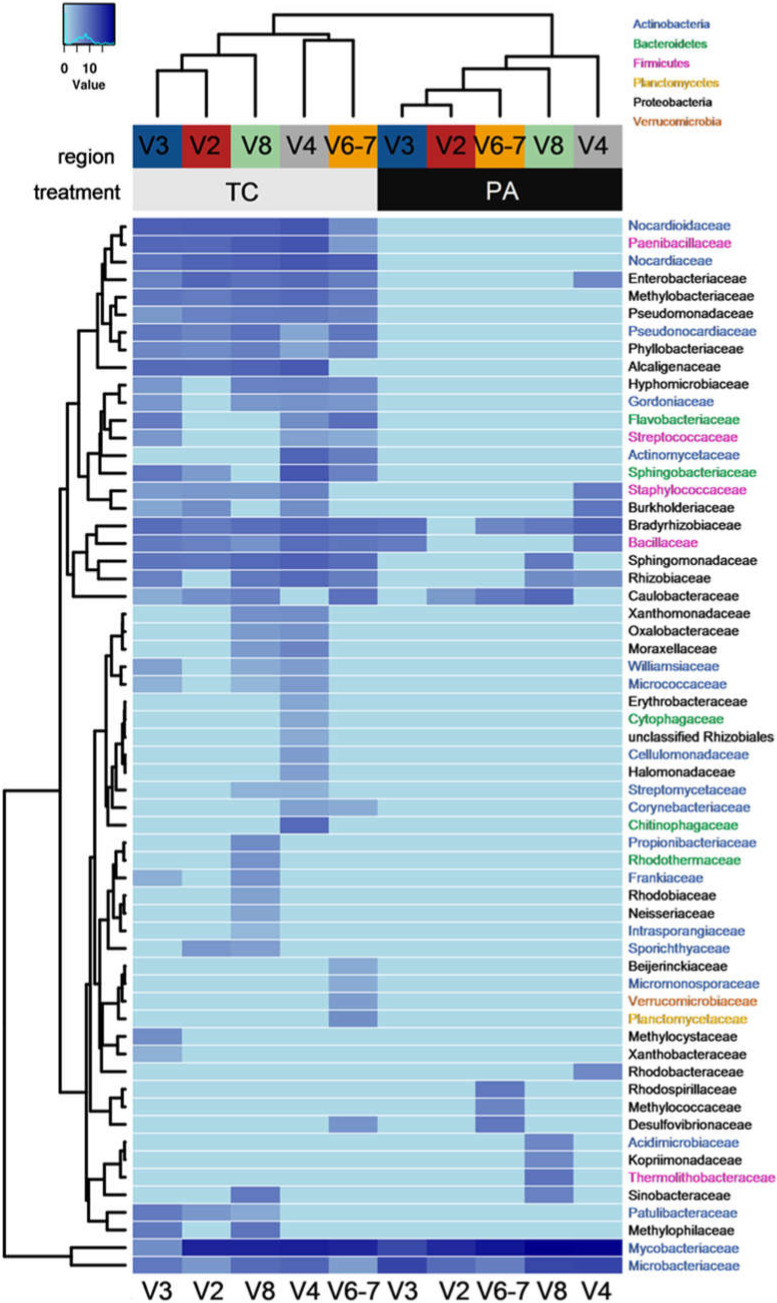
Family-level distribution of reads mapped to high-abundance families generated using five variable 16S rRNA region-specific primer pairs for the control (TC) and post-antibiotic treatment (PA) samples of the tobacco in vitro shoots. Data shown in Appendix A. Different font colors represent classes of the Bacteria domain.

**Figure 6 plants-11-00832-f006:**
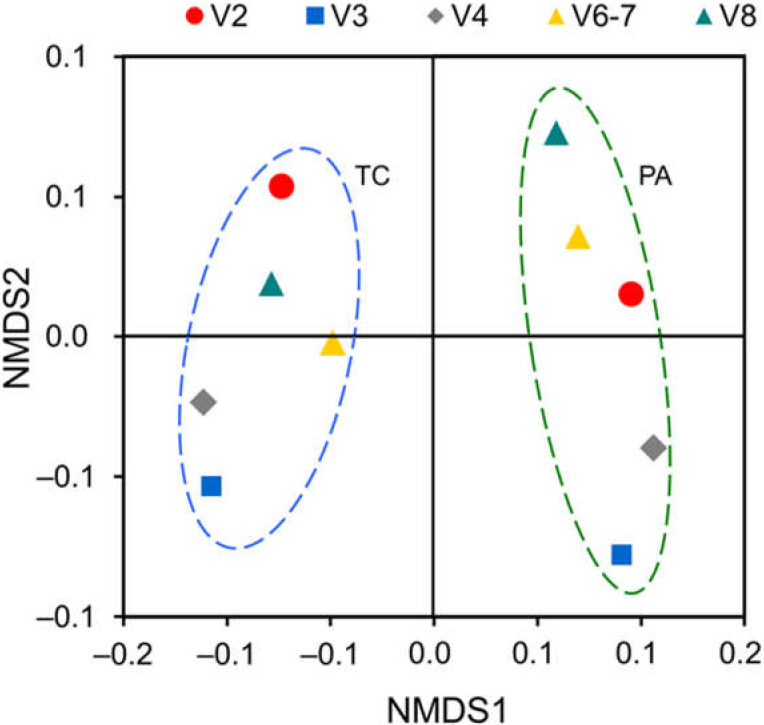
Nonmetric multidimensional scaling (NMDS) analysis of variability among the bacterial OTU data sets generated for the tobacco shoot control (TC) and post-antibiotic treatment (PA) samples using five 16S rRNA gene variable region-specific primer pairs. The analysis was carried out using the Bray–Curtis dissimilarity matrix and the OTU data of five primer pairs for each experimental group.

**Figure 7 plants-11-00832-f007:**
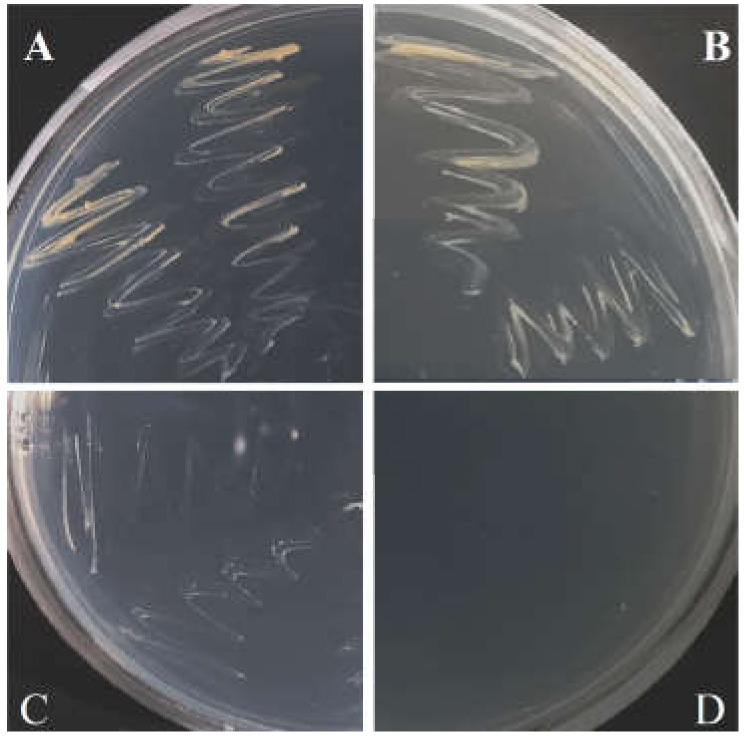
Growth of the isolate of *Mycobacterium* sp. related to the *M. cookii* strain ATCC 49103 cultivated for 30 days on Actinobacteria Isolation Agar (**A**), medium supplemented with 250 mg L^−1^ timentin (**B**), 30 mg L^−1^ chloramphenicol (**C**), or 25 mg L^−1^ rifampicin (**D**).

**Figure 8 plants-11-00832-f008:**
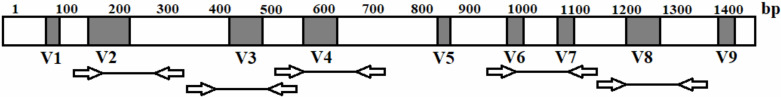
Five regions of the 16S rRNA amplified using V2, V3, V4, V6–7, and V8 domain-specific primers from the 16S Metagenomic kit (Thermofisher Scientific) used for multi-variable region-based 16S rRNA gene high-throughput sequencing analysis.

## Data Availability

Not applicable.

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
