# Peer review of "Enduring Effect of Antibiotic Timentin Treatment on Tobacco In Vitro Shoot Growth and Microbiome Diversity"

_plants, 2022, doi:10.3390/plants11060832_

Round 1

Reviewer 1 Report

Dear Authors,

Can the functional diversity of the identified bacteria be described in more detail? Do they promote plant growth, e.g., by producing auxin, and if they disappear due to antibiotic use, does this negatively affect further plant growth in in vitro cultures? What is the best level of microbiome biodiversity?
Do plants need to rebuild their specific microbiome after planting in soil? Have individual tobacco plants been observed to have the same or similar microbiome as opposed to other plant species? In light of the research, can we conclude that the use of antibiotics in crop protection has negative effects on endophytes (or rather endotrophs, since they are not part of the plant). In the past, a similar term was used for fungi, "saprophytes," but since the kingdom of "fungi" was created, the more correct term is "saprotrophs" or "saprobes." I think the same applies to the word "endophytes", which is better replaced by "endotrophs".
What is the importance of the dominant families Mycobacteriaceae and Microbacteriaceae, and the alpha-proteobacteria, Caulobacteriaceae, and Bradyrhizobiaceae? Which bacteria relieve stress? Are bacteria that are pathogenic to humans and animals also detected? What may be the consequences of the observed resistance of bacteria to antibiotics and the dominance of a few families? Should different antibiotics be alternated to prevent this? Should their use be abandoned by using other forms of sterilisation of plant material?

Author Response

We thank Reviewer for comments and insightful suggestions. Changes described below are indicated by Markup in the uploaded revised manuscript. Line numbers separated by slash correspond to line numbers used in the manuscript file with Markup and the revised manuscript without Markup, respectively.

  1. Can the functional diversity of the identified bacteria be described in more detail? Do they promote plant growth, e.g., by producing auxin, and if they disappear due to antibiotic use, does this negatively affect further plant growth in in vitro cultures? What is the best level of microbiome biodiversity?

These are important questions asked by the Reviewer. To address the Reviewer's comment on the functional diversity of the identified bacteria, we included a short overview of several taxonomic groups with potential growth-promoting traits in the Discussion (lines 399-401 and 418-436 in the file with Markup or lines 370-372 and 389-397 in the revised manuscript without markup). However, we would like to note that the microbiome analysis method used in our study is based on short-read sequencing of the 16S rRNA gene which could provide reliable taxonomic resolution mainly limited to the family level. This does not provide sufficient depth to identify functional aspects of the microbial community, such as phytohormone production or antibiotic resistance, as these metabolic traits are commonly specific to species or often strains of microorganisms. Therefore, to avoid excessive speculation, we would not like to expand on the potential functions of the identified taxonomic groups in the discussion. As indicated in the Discussion (line 436-439/397-400), refining the functions of the plant microbiome involved in the modulation of in vitro culture growth and adaptation would require further studies, potentially including specific bacterial isolates or microbial consortia.

  1. Do plants need to rebuild their specific microbiome after planting in soil? Have individual tobacco plants been observed to have the same or similar microbiome as opposed to other plant species? In light of the research, can we conclude that the use of antibiotics in crop protection has negative effects on endophytes (or rather endotrophs, since they are not part of the plant). In the past, a similar term was used for fungi, "saprophytes," but since the kingdom of "fungi" was created, the more correct term is "saprotrophs" or "saprobes." I think the same applies to the word "endophytes", which is better replaced by "endotrophs".

Again, these are all important questions to understand the role of the endophytic bacteria for plant growth and adaptation. However, we would like to note, that the main focus of our study was on the negative enduring effect of antibiotic timentin in the tobacco tissue culture and how it could affect the plant in vitro propagation efficiency. Our results may not provide a sufficient basis for broader discussion such as the effect on plant adaptation in soil or the effect of antibiotics used in crop protection, and these questions should potentially be addressed in future studies.

For the second part of the question, we agree that “endotroph” could be a more precise term to describe symbiotic or parasitic microorganisms that derive nutrients from within plant tissues. “Endophyte” is a more general term and commonly used to describe microorganisms that reside within plant tissues for at least part of their life cycle without causing apparent disease. As discussed in the response to comment No. 1, since our analysis is not capable to provide information about functional or metabolic interactions between identified bacteria and plant, we used a more general term of “endophyte”.

  1. What is the importance of the dominant families Mycobacteriaceae and Microbacteriaceae, and the alpha-proteobacteria, Caulobacteriaceae, and Bradyrhizobiaceae? Which bacteria relieve stress? Are bacteria that are pathogenic to humans and animals also detected? What may be the consequences of the observed resistance of bacteria to antibiotics and the dominance of a few families? Should different antibiotics be alternated to prevent this? Should their use be abandoned by using other forms of sterilisation of plant material?

As described in the response to comment No. 1, we included a short overview of several taxonomic groups with potential growth-promoting traits in the discussion. Specifically, the potential role of Mycobacteriaceae is addressed on lines 383-385/367-369. The identified Mycobacterium species isolate is an environmental bacterium that is not related to known human pathogens. As for other bacteria identified using 16S rRNA sequencing, the analysis is not capable to provide sufficient taxonomic resolution to identify specific human or animal pathogens, and we did not comment on this issue.

A potential connection between domination of bacterial families and antibiotic resistance is discussed in the paragraph including lines 389-398/360-369. Overall, it is a complex situation that depends on which bacteria acquire antibiotic resistance. For example, in case there plant beneficial bacteria would acquire antibiotic resistance, antibiotic treatment could be effective against certain pathogens and overall it might have a plant growth-promoting effect. Considering the complexity of the issue, we tried to avoid too general conclusions about the potential consequences of antibiotic use.

Reviewer 2 Report

Congratulations for the well-developed research and quality of the manuscript.

Author Response

We are grateful to Reviewer for the positive review of the manuscript.

Reviewer 3 Report

Dear Authors,
I have found your work, entitled: "Enduring effect of antibiotic timentin treatment on tobacco in vitro shoot growth and microbiome diversity" very interesting. You have done extensive work of importance in my opinion and I think your described information and conclusions could interest many researchers and readers. The thesis is a very wide compendium of knowledge about microbiome diversity in vitro culture, which is a very important issue.   
What is more, You created an extensive literature collection in this area, which is very beneficial for those interested in this topic. Although the work is interesting, I think that You should take a count modification of this article. I recommend publishing it in "Plants" after a small transformation. The only objection is the feeling that the Authors are mixing parts of the article with each other, eg in some of the results, there are fragments that should be included in the introduction or in the discussion, eg Lines 90-96 and others.

With best regards

Author Response

We are grateful to Reviewer for the positive review of the manuscript and comments. Changes described below are indicated by Markup in the uploaded revised manuscript. Line numbers separated by slash correspond to line numbers used in the manuscript file with Markup and the revised manuscript without Markup, respectively.

The only objection is the feeling that the Authors are mixing parts of the article with each other, eg in some of the results, there are fragments that should be included in the introduction or in the discussion, eg Lines 90-96 and others.

According to the Reviewer suggestion, paragraph including lines from 90-96 of the submitted manuscript were revised and partially moved to Introduction (lines 45-47) and Discussion (lines 373-378 /344-349) sections.

The paragraph of the Results section including lines 141-147 of the submitted manuscript was moved to the Materials and Methods section (lines 477-480 /438-441 of the revised manuscript).

Reviewer 4 Report

The manuscript entitled „Enduring effect of antibiotic timentin treatment on tobacco in vitro shoot growth and microbiome diversity“is very well written. The aim of the study was to assess the effect of antibiotic treatment on the growth and stress level of tobacco (Nicotiana tabacum L.) shoots in vitro as well as the composition of the plant-associated microbiome. In introduction the importance of the study is expressed in an understandable manner. The authors used 16S rRNA gene sequqncing-based analysis to investigate bacterial diversity in tobacco in vitro shoots. The antibiotic treatment induced decline in taxonomic diversity. The authors proposed that antibiotic-induced perturbation of shoot microbiome composition might contribute to the reduced adaptive capacity and impede the growth of tobacco shoots, leading to the reduced efficiency of in vitro culture propagation.

I suggest minor revision of the manuscript.

In Results and Discussion

Page 4  from line 141 „Six different DNA extraction an…………to line 147 is more suitable for Materials and Methods

Page 5 Figure 3 A

 Write correct values of proportion of bacterial sequences for the A and B DNA extraction methods of TC and PA samples.

Table S1 is not available in supplementary materials.

Author Response

We thank Reviewer for the suggestions on how to improve the manuscript. Changes described below are indicated by Markup in the uploaded revised manuscript. Line numbers are indicated as used in the manuscript file with Markup and the revised manuscript without Markup, respectively.

  1. In Results and Discussion. Page 4 from line 141 „Six different DNA extraction an…………to line 147 is more suitable for Materials and Methods

 According to the Reviewer suggestion, the text in the indicated paragraph on page 4 was moved to the Materials and Methods section (lines 477-480 /438-441).

  1. Page 5 Figure 3 A. Write correct values of proportion of bacterial sequences for the A and B DNA extraction methods of TC and PA samples.

 The values of the proportion of bacterial sequences are estimated based on the total number of mapped reads for each individual sample. Since the total number of mapped reads varies up to several times from sample to sample, the relative proportion of bacterial sequences may appear larger for samples with a lower total read number. The figure legend was supplemented on lines 195-196/167-168 to clarify this issue:

 “Numbers indicate the percent value of reads mapped to bacterial sequences estimated based on the total number of mapped reads for each individual sample.”

  1. Table S1 is not available in supplementary materials.

 Table S1 is available as Supplementary materials uploaded in a separate MS Excel file.

Reviewer 5 Report

  1. The logic of experimental procedure is a concern of this manuscript. Only one antibiotic and one concentration is used to test the inhibition effect on microbial growth. Besides, in general the first thing is to identify what microbes exist in the plant tissue and search which antibiotic can inhibit the growth or kill the microbe. In stead, the paper has a backward procedure.
  2. Most researchers already knew the effect of antibiotic on the growth of plant tissues, the experimental design for this part is rather weak as no other antibiotic and different concentration are compared. One reference for antibiotic concentration effect is as: https://doi.org/10.21273/HORTSCI.42.3.629
  3. Otherwise the manuscript is acceptable after some explanation.

Author Response

We thank Reviewer for suggestions and productive discussion of the manuscript. Changes described below are indicated by Markup in the uploaded revised manuscript. Line numbers are indicated as used in the manuscript file with Markup and the revised manuscript without Markup, respectively.

  1. The logic of experimental procedure is a concern of this manuscript. Only one antibiotic and one concentration is used to test the inhibition effect on microbial growth. Besides, in general the first thing is to identify what microbes exist in the plant tissue and search which antibiotic can inhibit the growth or kill the microbe. In stead, the paper has a backward procedure.

 As indicated in the title and objective of our study, the main focus of the study was on the negative enduring effect of antibiotic timentin and its consequences on the microbiome composition of the tobacco tissue culture. The selection of the antibiotic was mainly guided by the fact that timentin is considered relatively safe for plant tissue culture and is a very common choice for the Agrobacterium removal procedure used for plant transformation. To clarify this issue the description of the aim of the study was modified on lines 82-85:

“Therefore, the aim of this study was to assess the effect of antibiotic timentin, commonly used following Agrobacterium-mediated transformation, on the growth and stress level of tobacco shoots in vitro and the composition of the plant-associated microbiome.”

Further, our study revealed an enduring negative effect of the antibiotic on tobacco in vitro shoot growth that was detected after transfer to the medium without antibiotic and was observed for several passages used in the study. Considering that this enduring effect of the antibiotic treatment could not be explained by the direct cytotoxic effect of the antibiotic, we proposed that the antibiotic-induced perturbations in composition and/or interactions within the plant-associated microbial community of the tobacco shoots could be a significant contributing factor. Therefore, we assessed the antibiotic effect on the microbiome composition of tobacco in vitro shoots. The potential contribution of the antibiotic-induced perturbations in composition and/or interactions within the plant-associated microbial community of the tobacco shoots to the enduring negative effect on the growth and adaptive capacity of the plant tissue culture are discussed on lines (392 – 412/363-383) of the Discussion.

To address the second part of the Reviewer comment, we have to point out that we first analyzed the microbiome in tobacco in vitro shoots without treatment and then evaluated how that microbe composition was affected by antibiotic treatment.

  1. Most researchers already knew the effect of antibiotic on the growth of plant tissues, the experimental design for this part is rather weak as no other antibiotic and different concentration are compared. One reference for antibiotic concentration effect is as: https://doi.org/10.21273/HORTSCI.42.3.629

We agree that the cytotoxic effect of antibiotics in tissue culture is known fact and has been previously described. This problem was addressed on lines 45-47 of the Introduction and lines 373-374 /344-345 of the Discussion section. There we also included a reference to the study by Chen and Yeh (2007) suggested by Reviewer.

However, as indicated in the response to Reviewer comment No. 1, the main focus of our study was not on the direct cytotoxic effect of antibiotic timentin but on the growth suppressing enduring effect of the antibiotic. A low cytotoxic effect of the antibiotic timentin on in vitro plant tissues had been documented previously and it is commonly used following Agrobacterium-mediated transformation. However, our experiments showed suppressed growth of tobacco in vitro shoots on a medium supplemented with timentin and remarkably a consistent residual negative effect of antibiotic treatment on shoot growth vigor and stress level was observed for at least several passages after transfer to the medium without antibiotic. The antibiotic-induced perturbations in composition within the plant tissue culture microbiome were addressed as described in the response to Reviewer comment No. 1.

  1. Otherwise the manuscript is acceptable after some explanation.

 We hope the explanation included in the response to the Reviewer comments No. 1 and 2 and the manuscript text revised according to Reviewer suggestions provide sufficient clarification of the study objectives for the Reviewer and potential readers.